# Pore Evolution Characteristics of Marine Organic-Rich Shale Based on a Pyrolysis Simulation Experiment

**Xin Zhao \*, Wen Zhou** **, Hao Xu, Wenling Chen and Ke Jiang**

State Key Laboratory of Oil and Gas Reservoir Geology and Exploitation, Chengdu University of Technology, Chengdu 610059, China
* Correspondence: zhaoxin@stu.cdut.edu.cn

**Abstract:** The pore evolution characteristics of organic-rich shale under semiclosed and open-system conditions were investigated through a pyrolysis simulation experiment conducted on marine low-maturity shale. The results show that the experimental samples of the two systems mainly developed organic pores, clay mineral intergranular pores, and brittle mineral dissolution pores during the evolution process. The shale porosity showed a non-linear growth trend with the thermal maturity increase. However, the porosity increase rate of the semiclosed system was higher than that of the open system. The pore structure characterization parameters of the experimental samples under the two systems were quite different. The samples of the semiclosed system developed wedge- or V-shaped pores with a diameter of several tens of nanometers, while those of the open system developed more small throats with a diameter of several nanometers to ink bottle-type and closed pores of tens of nanometers. For the marine organic-rich shale, the organic matter (OM) evolution was mainly affected by the formation pressure. In summary, a semiclosed system with a higher formation pressure is more conducive to the development of organic pores and the pore structure than an open system with a lower formation pressure. The best stage of pore evolution and development is the overmature stage.

**Keywords:** Dalong Formation; marine shale; pore evolution; thermal simulation experiment; pore structure characteristics; semiclosed system; open system

## 1. Introduction

Unlike those in conventional sandstone reservoirs, the pores of shale reservoirs are characterized by a complex structure and a strong heterogeneity. Additionally, their evolution is controlled by many factors that cannot be characterized and evaluated by conventional reservoir pore research methods. Therefore, studying the pore structure, genetic mechanism, thermal evolution characteristics, and influencing factors of shale reservoirs is important for the exploration and development of shale gas. Previous studies have shown that compared with inorganic pores, organic pores in shale are an important storage space for hydrocarbon gas in shale reservoirs. Hence, the evolution characteristics of organic pores are extremely important for shale reservoir evaluation [1–6].

At present, research on the evolution of the organic matter pore structure of shale reservoirs usually employs outcrop or well samples in different maturities and thermal evolution simulation experiments of low-evolution samples [4–7]. Experiments are an effective approach for exploring the evolution law of shale pores. The thermal evolution experiment mainly studies the dynamic process of the formation and development of shale pores, which is based on the principle of the thermal degradation and the principle of the time–temperature compensation [6,7]. Thermal experiments conducted in laboratories select immature or low-maturity shale and use high temperature conditions to compensate for the geological effects of time and pressure on organic matter under actual geological conditions and approach the actual formation condition effects on the organic matter evolution.

Previous scholars have used thermal evolution experiments to analyze the hydrocarbon generation potential and hydrocarbon components of shale; however, there is less research on the evolution characteristics of mud shale pores, and relevant experimental research on the development degree of shale pores in different systems is lacking. In addition, there are also many imprecisions in the experimental design of thermal evolution [8–14]. These experiments cannot accurately reflect the pore evolution characteristics and the porosity changes of source rocks in the actual stratigraphic environment.

In this paper, outcrop shale samples with low thermal maturity from Dalong Formation in Changjianggou, Guangyuan (Sichuan Basin) is collected. Thermal experiments in two different systems (i.e., semiclosed and open) are then conducted, and their results are compared. This work aims to characterize the pore structure (i.e., pore type, pore size distribution, and pore volume) at different thermal maturity levels, reveal the pore evolution during the thermal maturity from low to extremely high thermal maturity levels, and discuss the main controlling factors for the pore generation and preservation of organic-rich shales.

*Samples and Experiment*

The Upper Permian Dalong Formation ($P^2_3dl$) is locally deposited in the Upper Yangtze region, a contemporaneous heterogeneous deposition of the Upper Permian Changxing Formation ($P^2_3ch$). Compared to other marine source rocks, Dalong Formation is mainly distributed in Longmen and Micang mountains, Guangyuan, Wangcang, Chengkou, Wushan, and areas around the northwestern Sichuan Basin, with a thickness varying from a few meters to tens of meters. The thermal maturity degree of the Dalong Formation shale in different regions is also quite different. The Dalong Formation maturity in most areas of the Sichuan Basin has reached the mature or high-mature stage, with an equivalent thermal maturity between 0.52% and 1.78% [14–17]. The low thermal maturity of Dalong Formation can be found in the Guangyuan area. The equivalent vitrinite reflectance (Ro) of the source rocks is approximately 0.7%. The kerogen type of Dalong Formation is I-II$_1$. Dalong Formation mainly comprises microcrystalline limestone and siliceous shale, which are in conformity with the limestone of the underlying Wujiaping Formation and the silty mudstone of the overlying Feixianguan Formation. The mineral components of the collected shale sample are mainly quartz, calcite, and clay minerals with a small amount of plagioclase, dolomite, and pyrite. The brittle mineral content is high. The main components of the clay minerals are illite and illite mixed layers [18,19].

Outcrop samples were taken from Dalong Formation in the Changjianggou section of Guangyuan in the north Sichuan Basin (Figure 1). Dalong Formation in the Changjianggou section mainly comprises black medium thin-layer calcareous shale and gray-black-gray middle-layer silicon limestone. Thermal evolution experiments require low-evolution organic-rich samples; hence, to reduce the impact of shale heterogeneity, the column samples must be drilled from the same sample. Accordingly, one black shale with a larger volume, a lower maturity, and a higher organic carbon content was selected as the experimental sample. Preliminary experiments, including TOC determination, Ro, and X-ray diffraction, were performed on the as-received rock samples. CJG-2 was also finally selected as an experimental sample (Table 1).

**Table 1.** Sample data.

| Number | Mineral Content (%) | | | | | | | | Ro (%) | TOC (%) |
|---|---|---|---|---|---|---|---|---|---|---|
| | Quartz (Q) | Plagioclase | Calcite (Cal) | Dolomite | Pyrite (Py) | Clay | Imon Mixed Layer | Illite | | |
| CJG-2 | 34 | 4 | 24 | 4 | 4 | 30 | 69 | 31 | 0.5 | 9.14 |

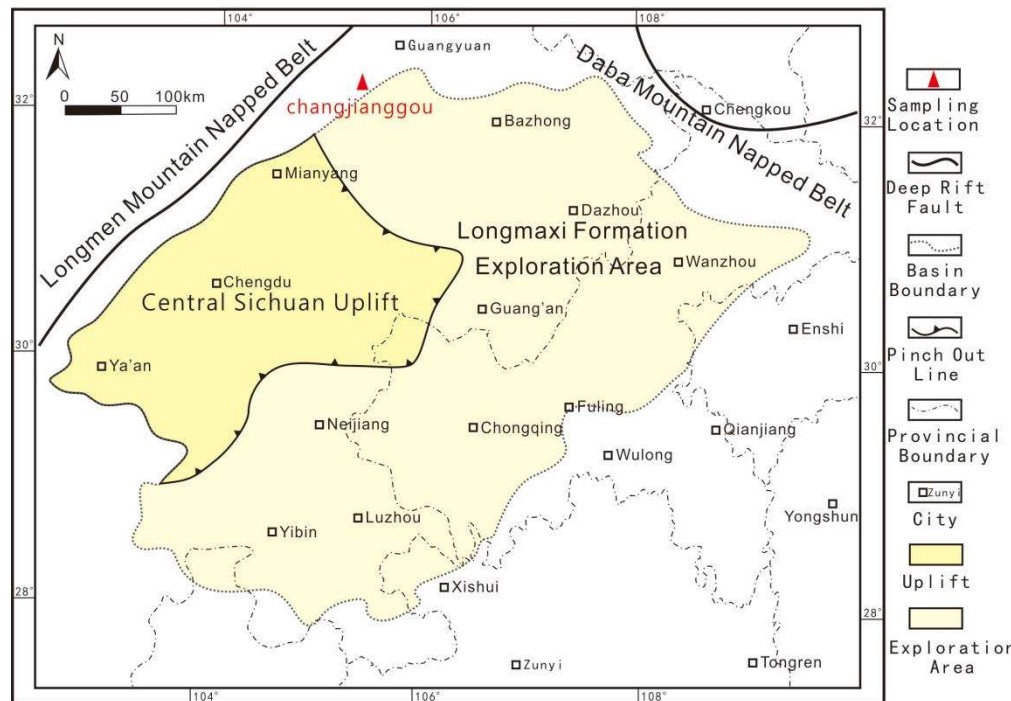

**Figure 1.** Sketch map of Sichuan Basin and the sampling location.

A pyrolysis simulation experiment was conducted on the formation porosity thermo-compression simulation experiment instrument at the Wuxi Petroleum Geology Research Institute of Sinopec Petroleum Exploration and Development Research Institute [20]. Two groups of experiments using the semiclosed (Group 1) and open (Group 2) systems were conducted. Their results were then compared. The semiclosed system ensures a certain formation pressure in the system and starts to expel hydrocarbons after the formation pressure is exceeded. Meanwhile, the open system almost has no formation pressure in it, and the generated hydrocarbon is simultaneously expelled at any time. The experimental flow is described as follows:

(1) Twelve column samples were divided into two groups (i.e., six for a group) and placed in the instrument. Brine with 40 mg/L salinity was used as the pore fluid.
(2) The static rock pressure was set to maintain a certain pressure difference between the two groups of static rock pressure and formation pressure (Table 2).
(3) The target temperature was set to 350 °C, and the heating rate was 20 °C/h. The temperature was sustained for 2 days after reaching the target temperature. The fluid pressure was then checked and naturally cooled to room temperature. The samples after the pyrolysis simulation experiment were taken out for a scanning electron microscope observation and a series of follow-up experiments, including TOC determination, gas composition analysis, X-ray diffraction whole rock analysis, and clay mineral determination analysis.
(4) The second sample was replaced, the target temperature was set to 400 °C. Step (3) was then repeated.
(5) The temperature was set to 450 °C, 500 °C, 550 °C, and 600 °C. Step (3) and (4) were then repeated.

SEM analysis is usually applied to observe the pore types and pore abundance in samples. In the experiment, a field emission environmental scanning electron microscope (Quanta250 FEG, FEI company, Hillsboro, OR, USA) was used to describe the pore types and evolution. The samples were pieced in 1 cm² and milled in GATAN685 (GATAN company, Pleasanton, CA, USA) with a 30 kV focused argon ion beam. They were then coated with a Au alloy before the SEM analysis.

**Table 2.** Experimental temperature and pressure used for the pyrolysis simulation experiment.

| Number | | Temperature °C | Formation Pressure MPa | Static Rock Pressure MPa | Ro % | TOC % |
|---|---|---|---|---|---|---|
| Group 1 | CJGGY-1-350 | 350 | 37 | 89 | 1.28 | 1.41 |
| | CJGGY-1-400 | 400 | 50 | 125 | 1.73 | 1.4 |
| | CJGGY-1-450 | 450 | 59 | 148 | 2.31 | 1.24 |
| | CJGGY-1-500 | 500 | 61 | 153 | 2.78 | 1.11 |
| | CJGGY-1-550 | 550 | 71 | 176 | 3.29 | 1.15 |
| | CJGGY-1-600 | 600 | 79 | 196 | 3.49 | 1.08 |
| Group 2 | DYDB-1-350 | 350 | - | 89 | 1.45 | 0.99 |
| | DYDB-1-400 | 400 | - | 125 | 1.6 | 0.95 |
| | DYDB-1-450 | 450 | - | 148 | 1.55 | 0.94 |
| | DYDB-1-500 | 500 | - | 153 | 1.72 | 1.01 |
| | DYDB-1-550 | 550 | - | 176 | 1.76 | 0.98 |
| | DYDB-1-600 | 600 | - | 196 | 3.18 | 1.01 |

A series of experiments were conducted to characterize the pore system and the gas storage potential. The gas adsorption experiment was conducted on Micromeritics MicroActive for the ASAP 2460 2.02 surface area and porosity tester (Micromeritics Corporate, Norcross, GA, USA). During the experiment, the sample was pulverized to 40–100 mesh and placed into the instrument sample tube. The gas adsorbed on the pore surface was removed in a heated inert gas environment. The pressure to measure the corresponding value was then gradually changed. The adsorption and desorption tests were performed to obtain the adsorption-desorption isotherms of different samples. The nitrogen adsorption results were interpreted herein using the BJH model.

The instrument used in the mercury porosimetry experiment is Micromeritics Autopore 9500 mercury porosimeter (Micromeritics Corporate, Norcross, GA, USA). The sample was first dried, then mercury was injected into the sample tube under vacuum conditions. The test and the analyses were conducted in low (0–400 MPa) and high (1–227 MPa) pressures. In the low-pressure test, during which inactive drying gas (e.g., nitrogen) was injected, the pressure was continuously increased. When the maximum external pressure was reached, the pressure was reduced to atmospheric pressure, and the sample tube was moved to the high-pressure station for analysis. The pores were gradually filled with mercury to obtain the relationship between the indentation amount and the pressure.

## 2. Experiment Results

### 2.1. Pore Morphological Characteristics

The pores developed in the original samples were mainly inorganic pores, including intergranular pores and microcracks (Figure 2a,b). The organic matter was distributed between the brittle mineral particles and around the spheroidal pyrite or coexisted with the clay minerals. Only a few micropores were developed in the organic matter (Figure 2c). This situation is attributed to the fact that the initial sample was in the shallow burial stage, and the organic matter was affected by the biochemical action and the low-temperature thermal action to form a small amount of organic matter pores.

Inter, intra, and organic pores and cracks were mainly developed when the evolution stage reached the mature stage during the thermal evolution experiment. The pores in the shale were gradually developed as the temperature increased. Overall, the pore developments of the two systems were significantly different. The pore development of the semiclosed system was more significant than that of the open system.

#### 2.1.1. Organic Pore

Organic pores are secondary pores formed by the cracking of solid kerogen during thermal evolution. These are important spaces for hydrocarbon storage in shale reservoirs [21–27]. They have diverse types and complex origins. In some organic matter,

the pores are developed, while in others there are no pores. The thermal maturity of organic matter increases as the temperature increases. After kerogen enters the oil generation threshold, the organic matter is cracked to generate liquid hydrocarbons. Pores are then formed inside the organic matter. As the maturity continues to increase, the continuous shrinkage of organic matter makes the number of pores increase, and the pore size becomes larger. Finally, pores with a larger pore size and better connectivity are formed inside and at the edge of the organic matter.

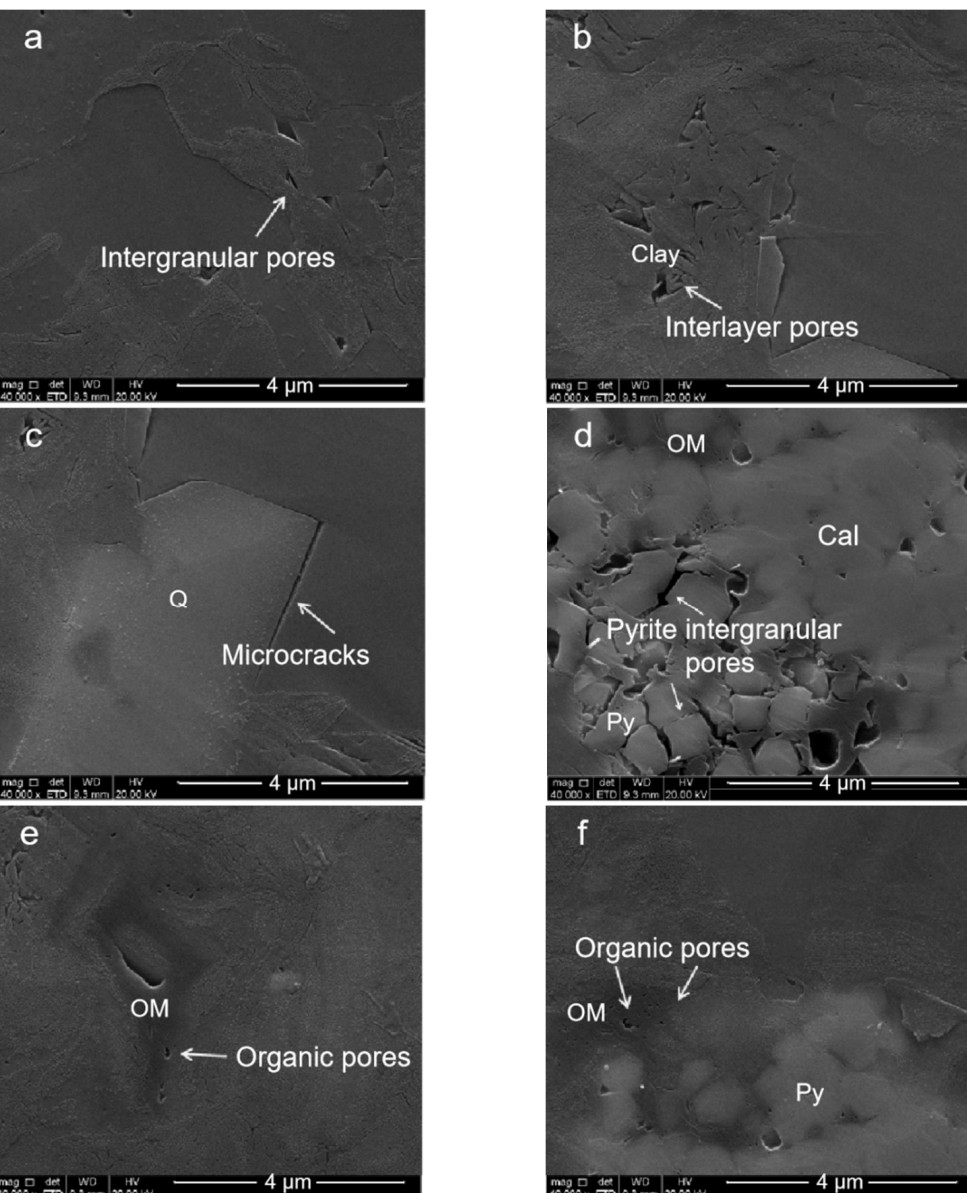

**Figure 2.** Original sample (Ro = 0.5%), where only a few micropores are developed: (**a**) intergranular pores; (**b**) clay interlayer pores; (**c**) microcracks; (**d**) pyrite intergranular pores; (**e**) organic pores; and (**f**) organic pores.

The pyrolysis simulation experiments were conducted on the same sample at six different temperatures. The corresponding maturity of shale varied from low to mature, to over mature. Due to the different sealing of the systems, the evolution degrees of the organic matter pores in the two systems were also very different at the same temperature (Figures 3 and 4).

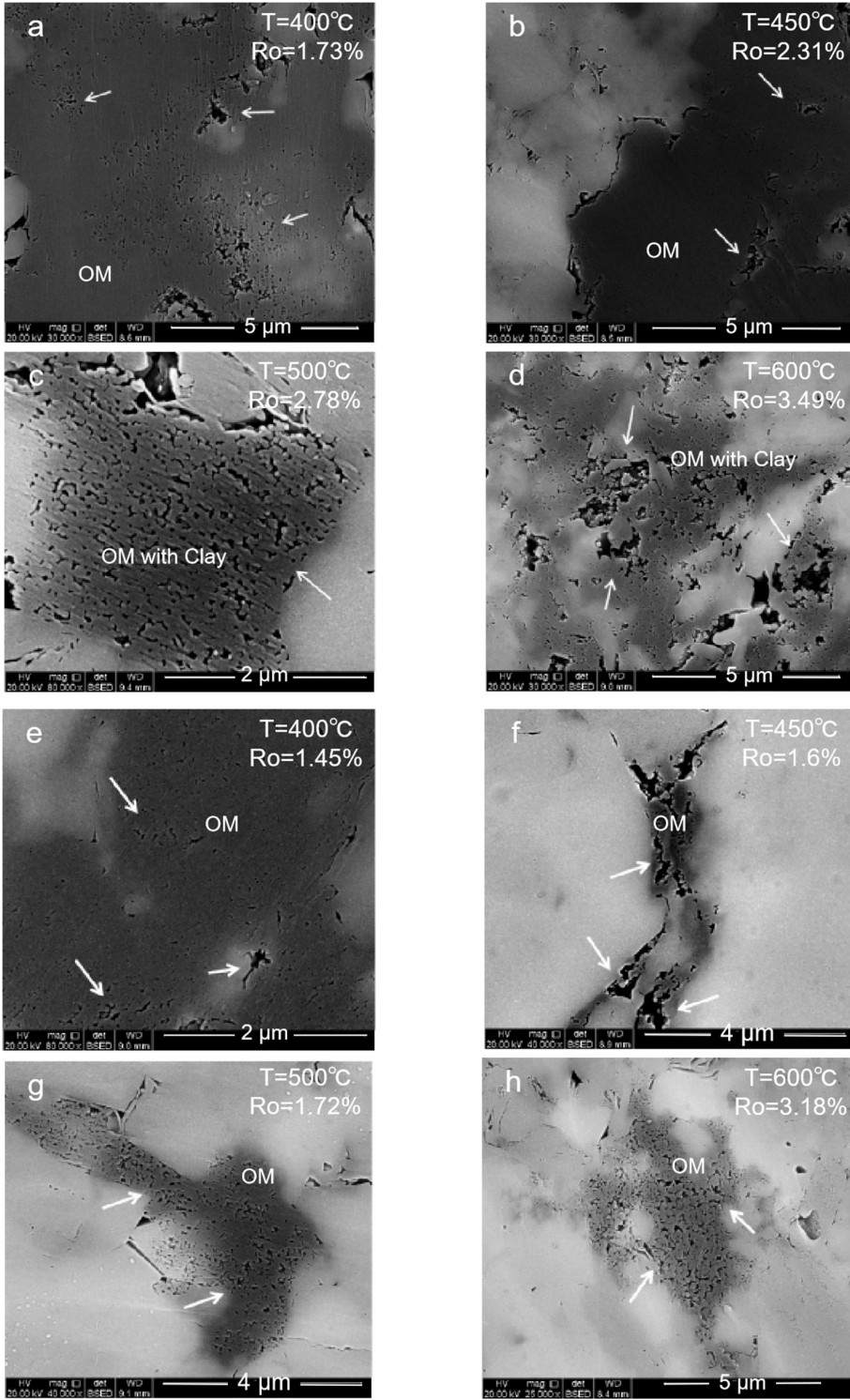

**Figure 3.** Evolution of the organic pores in the two groups under different temperatures. (**a**) Group 1, semiclosed system, T = 400 °C, organic matter begins to crack, bitumen fills the original pores. (**b**) Group 1, semiclosed system, T = 450 °C, organic pores further develop and enlarge. (**c**) Group 1, semiclosed system, T = 500 °C, a large number of shrinkage seams develop inside the organic matter. (**d**) Group 1, semiclosed system, T = 600 °C, the organic pores expand, connect, and decompose the organic matter framework. (**e**) Group 2, open system, T = 400 °C, a few organic pores are developed. (**f**) Group 2, open system, T = 450 °C, edge cracks and shrinkage cracks develop in the organic matter. (**g**) Group 2, open system, T = 500 °C, the shrinkage seams develop inside the organic matter. (**h**) Group 2, open system, T = 600 °C, organic pores are developed.

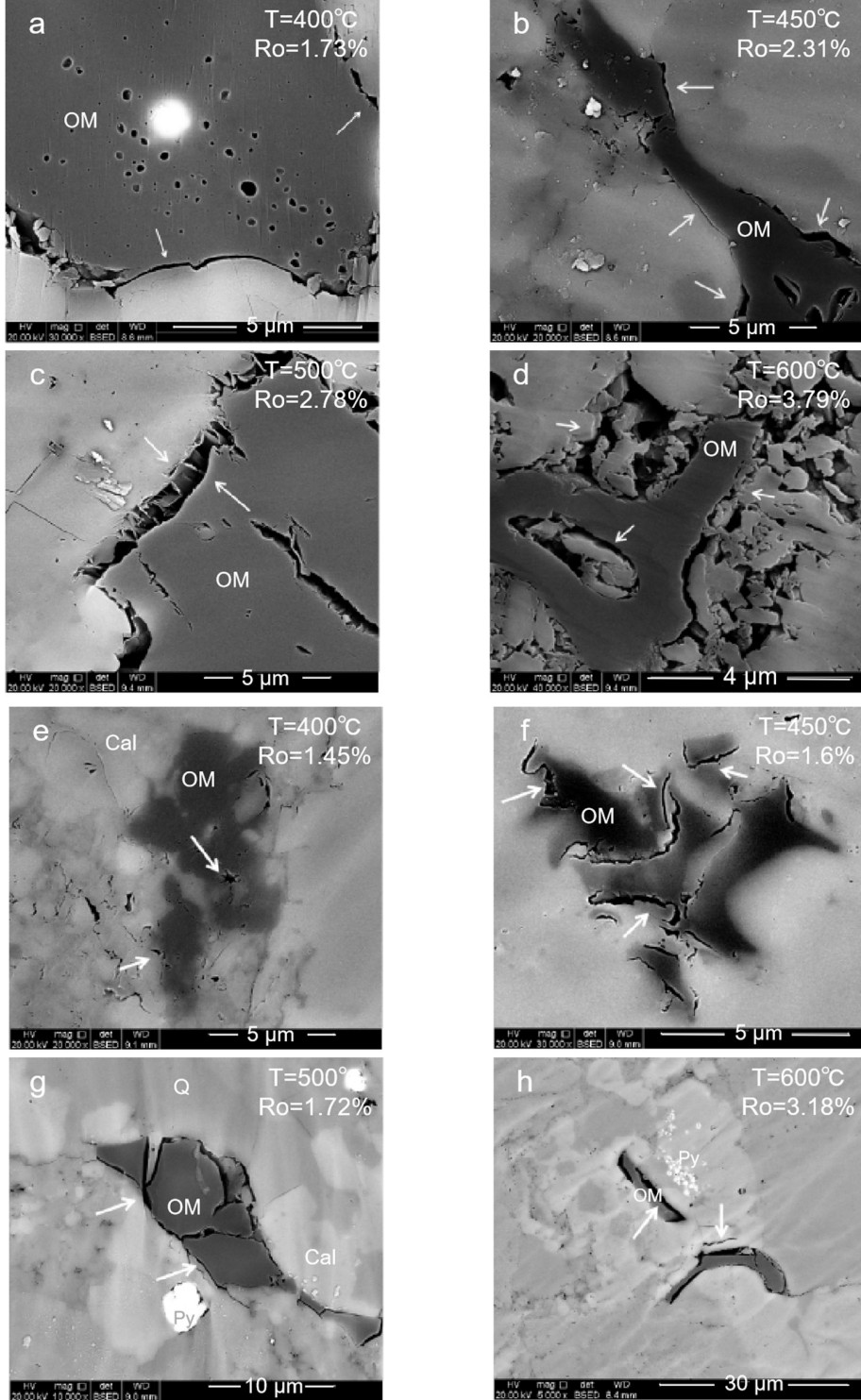

**Figure 4.** Evolution of the organic edge seam in two groups under different temperatures. (**a**) Group 1, semiclosed system, T = 400 °C, edge seams between the organic matter and the inorganic minerals. (**b**) Group 1, semiclosed system, T = 450 °C, edge seams enlarge. (**c**) Group 1, semiclosed system, T = 500 °C, organic inner shrink seam and organic edge seam. (**d**) Group 1, semiclosed system, T = 600 °C, organic matter edge seams and inorganic pores are connected. (**e**) Group 2, open system, T = 400 °C, organic edge seam was not developed. (**f**) Group 2, open system, T = 450 °C, the organic edge seam was not developed. (**g**) Group 2, open system, T = 500 °C, marginal and organic edge seams were developed in agglomerated organic matter. (**h**) Group 2, open system, T = 600 °C, the organic edge seam is squeezed and deformed.

In the semiclosed system, abundant pores were generated due to the formation pressure. When the temperature reached 350–400 °C, inorganic pores were still retained, and the organic matter began to crack to generate hydrocarbons. Subsequently, organic pores started to develop. At this stage, small circular or oval asphalt pores were formed in the asphalt (Figure 3a). Small, continuous, and dispersed organic matter stacking pores began to form. Some organic matter shrank, and shrinkage joints were formed at the edge of the organic matter–inorganic minerals (Figure 4a). At a temperature of 450 °C, the organic matter shrinkage joints and the asphalt pores further developed and enlarged. The organic matter accumulation pores also began to develop In large numbers. This time, the inorganic minerals were affected by the organic acid generated by the thermal evolution of the organic matter, resulting in some dissolution pores (Figures 3b and 4b). In the overmature stage at 500–600 °C, a large number of shrinkage seams developed inside the organic matter and in some edge seams. Some organic matter was divided into several pieces by these microfractures interconnected with the inorganic pores around the organic matter. The pores further expanded and connected and decomposed the organic matter framework (Figure 3c,d and Figure 4c,d).

In the open system, the organic matter evolution was stagnant because there was no formation pressure, and there were fewer visible organic pores. Some inorganic pores (e.g., interparticle and dissolution pores) were observed when the temperature reached 350–400 °C. The organic matter maturity did not obviously change with the increase in the temperature and the lithostatic pressure. Only a few edge cracks developed in and around the organic matter (Figures 3e–g and 4e–g). Abundant organic pores were generated and observed at 600 °C (Figures 3h and 4h).

### 2.1.2. Clay Mineral Pores

Compared with the original sample, the clay mineral components after the experiment are greatly changed. The illite content in the samples of the two systems was altered at 350 °C. No significant change was observed after that. The change was caused by the fact that during the thermal evolution, the clay minerals were transformed, and the mixed layer lost the interlayer water when heated and transformed into illite.

Compared to the original sample (Figure 5a), the shape of the clay mineral pores in the semiclosed system samples significantly changed. The pore size and the number also increased (Figure 5b). The relative content of the mixed layer decreased from 69% to less than 10%. The pore shape, structure, and composition gradually stabilized (Figure 5c). The pore deformation or closure occurred only in the overmature stage due to compaction (Figure 5d). In contrast, as for the open-system samples, the clay mineral pores began to develop at 450 °C and were affected by various maturity levels (Figure 5g).

The clay mineral transformation mainly occurred in the mature stage. In the high- and overmature stages, the main component of the clay minerals was illite. The pore size and structure tended to stabilize. Compared with the abundant organic micropores developed in the high- and overmature stages, the contribution of the clay mineral pores to the effective storage space of the reservoir was relatively limited.

### 2.1.3. Brittle Mineral Pores

After the organic matter entered the oil generation threshold, the fluid generated by the hydrocarbon generation changed the surrounding fluid environment, resulting in the dissolution pores developed in unstable brittle minerals, such as calcite and feldspar. Figure 6 shows many brittle mineral pores in the original sample (Figure 6a,e) and were dominated by dissolution pores and fractures. Calcite and feldspar around the organic matter must be dissolved significantly with the maturity increase (Figure 6b,f). However, no obvious pores were observed in quartz, and dissolution mainly occurred around the quartz grains (Figure 6c,g). The dissolution pores and the edge seams generated by the organic matter shrinkage were connected when the simulated temperature reached 550 °C (Figure 6d,h). It can be speculated that the brittle mineral dissolution pores could have

a great impact on the pore connectivity. A comparison of the pore development in the samples of the two systems revealed that the development degree of the brittle mineral dissolution pores in the samples of the semiclosed system was better than that of the open system.

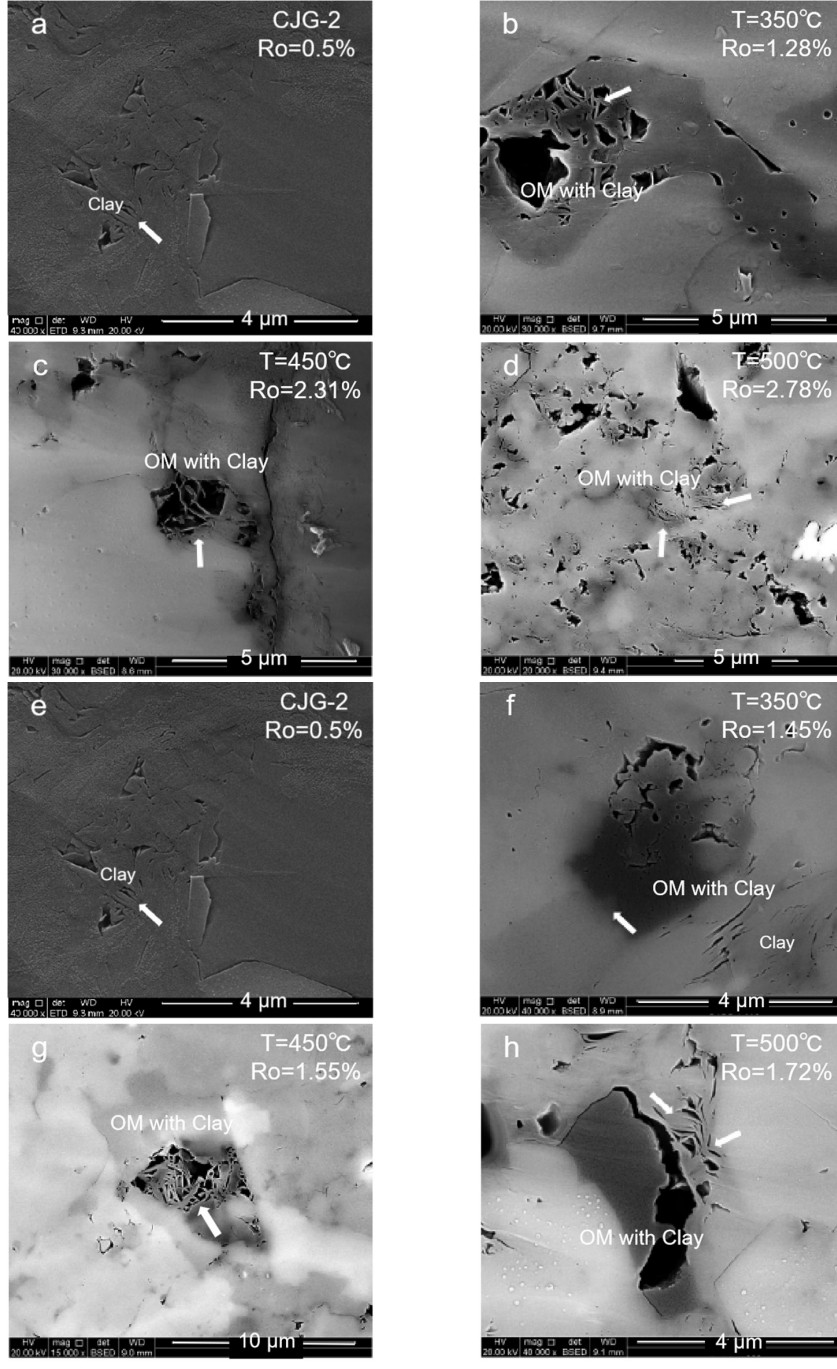

**Figure 5.** Evolution of the clay mineral pores in the two groups under different temperatures. (**a**) Original sample, little clay mineral pores were developed. (**b**) Group 1, semiclosed system, T = 350 °C, the clay mineral pores were enlarged. (**c**) Group 1, semiclosed system, T = 450 °C, no significant change in the clay mineral pores. (**d**) Group 1, semiclosed system, T = 500 °C, the clay mineral pores were squeezed. (**e**) Original sample, little clay mineral pores were developed. (**f**) Group 2, open system, T = 350 °C, the clay mineral pores were enlarged. (**g**) Group 2, open system, T = 450 °C, no significant change in the clay mineral pores. (**h**) Group 2, open system, T = 500 °C, the clay mineral pores were squeezed.

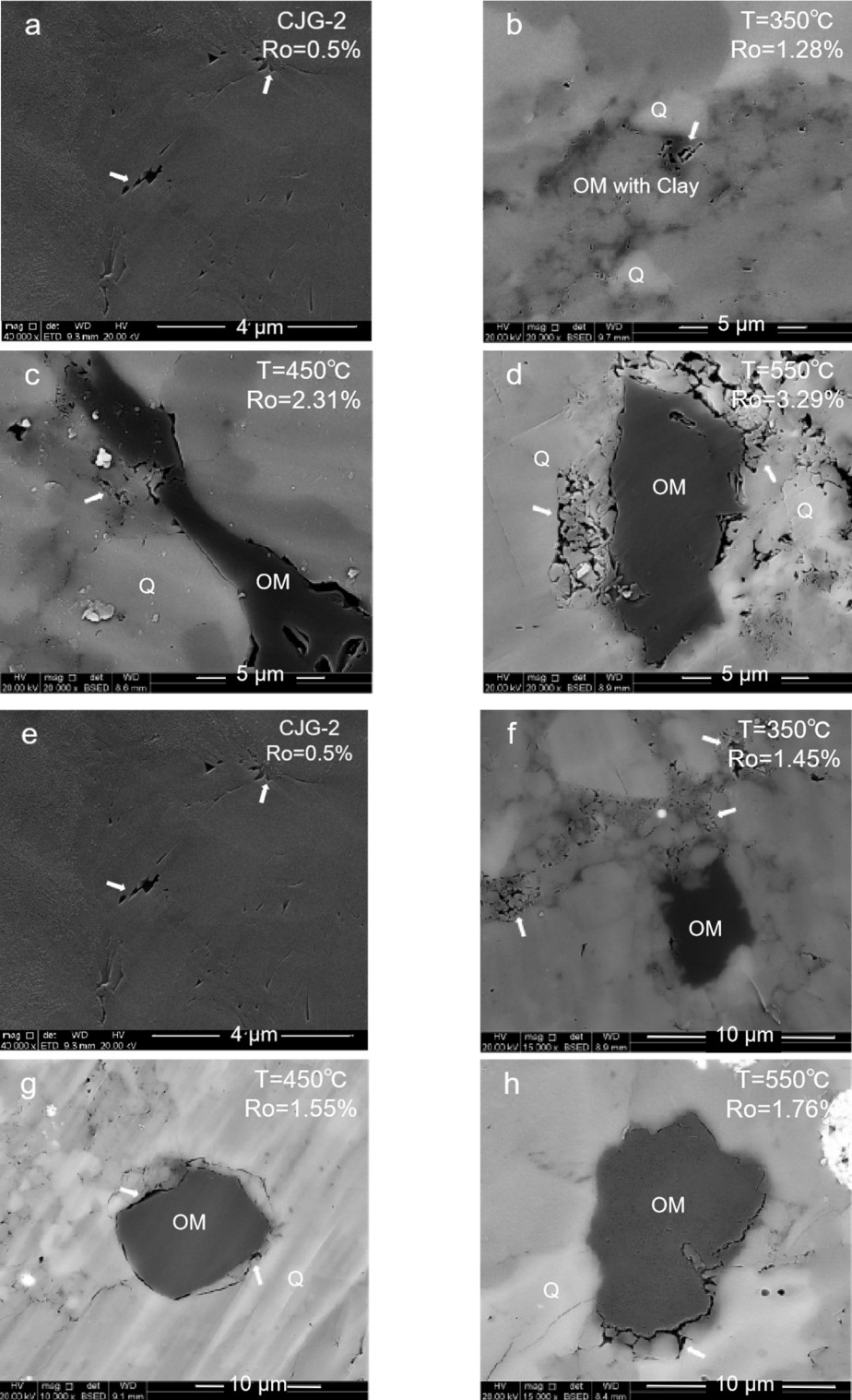

**Figure 6.** Evolution of the brittle mineral pores in two groups under different temperatures. (**a**) Original sample, many primary brittle mineral pores in the original sample. (**b**) Group 1, semiclosed system, T = 350 °C, little dissolution pores around the organic matter. (**c**) Group 1, semiclosed system, T = 450 °C, dissolution pores around the organic matter. (**d**) Group 1, semiclosed system, T = 550 °C, dissolution pores were enlarged. (**e**) Original sample, many primary brittle mineral pores in the original sample. (**f**) Group 2, open system, T = 350 °C, dissolution pores with clay mineral. (**g**) Group 2, open system, T = 450 °C, dissolution pores around the organic matter. (**h**) Group 2, open system, T = 550 °C, dissolution pores were enlarged.

*2.2. Pore Structure*

Shale mainly develops nanoscale pores. The pore size distribution, shape, and connectivity are extremely complex, which can greatly affect the effective porosity and permeability of the shale reservoir [25–33]. The pore structure, including the pore–throat distribution, specific surface area, average pore size, and porosity of the samples, were characterized by combinations of alcohol method porosity, low-temperature nitrogen adsorption, and high-pressure mercury intrusion experiments.

The shape of the mercury injection curve can reflect the development of the reservoir pore throats and the pore connectivity [34,35]. Figure 7 shows that the mercury intrusion curves of the two systems are quite different. As for the samples of the semiclosed system, the mercury injection saturation and withdrawal efficiency showed an overall upward trend with the temperature increases. This result indicates the increase in pore–throat size and a better pore connectivity. A higher mercury withdrawal efficiency at higher maturity levels indicates better pore connectivity, which is beneficial to the accumulation and migration of hydrocarbons. In contrast, the open-system samples had extremely low mercury injection saturation and withdrawal efficiency, suggesting smaller pore throats and a poor pore connectivity.

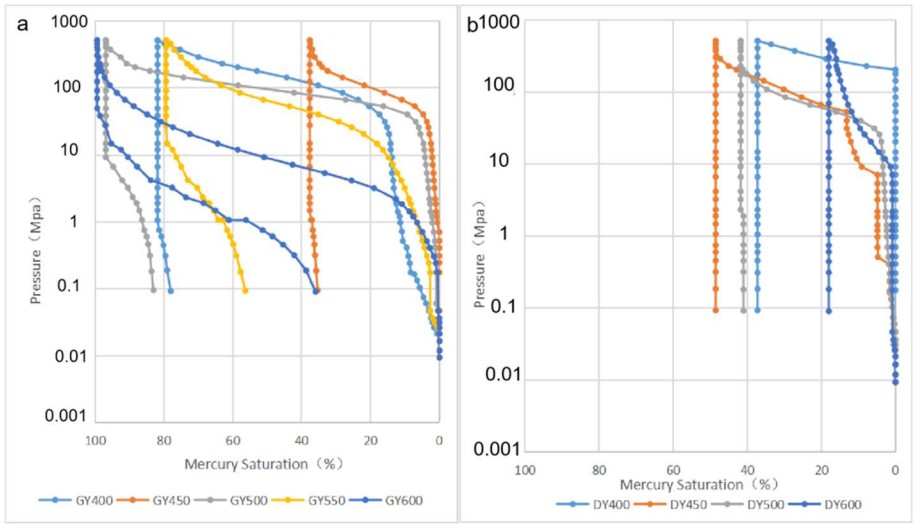

**Figure 7.** Mercury intrusion experiment curve. (**a**) Semiclosed and (**b**) open systems.

Comparing the mercury intrusion curve of GY600 with Longmaxi formation samples (Figure 8), it can be found that in the similar maturity, the mercury injection curves of GY600 and Longmaxi samples are both increased slowly with the increase in pressure at first, then they increased rapidly. Additionally, GY600 has a wide hysteresis loop between the injection curve and the withdrawal curve, the residual mercury saturation reaches 35.96%, while type I is 7.31% and type II is 58.18%. These features show that GY600 has a developed pore structure, and they are similar in pore development.

The pore size distribution obtained from the mercury intrusion experiment (Figure 7) showed that micropores and mesopores were developed during the evolution in the two systems. The mesopores or macropores were dominant pores. The development of the macropores with a pore size of more than a few hundred nanometers was limited and almost invisible after the evolution reached the overmature stage. Therefore, the thermal evolution process of organic matter is conducive to the development of micropores to mesopores and has a certain inhibitory effect on the development of macropores larger than several hundred nanometers.

After the evolution to the high-mature and overmature stage, the semiclosed system mainly developed mesopores and macropores (Figure 9a), while the open system mainly

developed micropores and mesopores (Figure 9b). The pore development in the semiclosed and open systems were quite different.

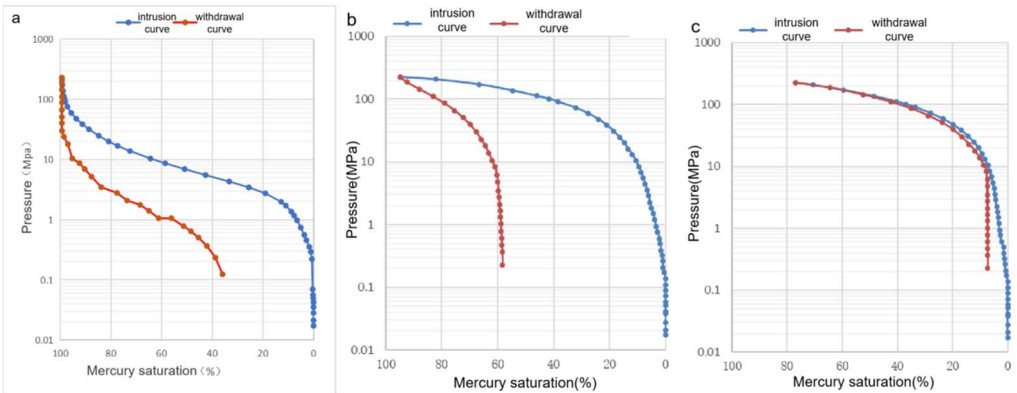

**Figure 8.** Mercury intrusion experiment curves. (**a**) GY600 of semicolsed system (**b**) Longmaxi formation type II and (**c**) Longmaxi Formation type I.

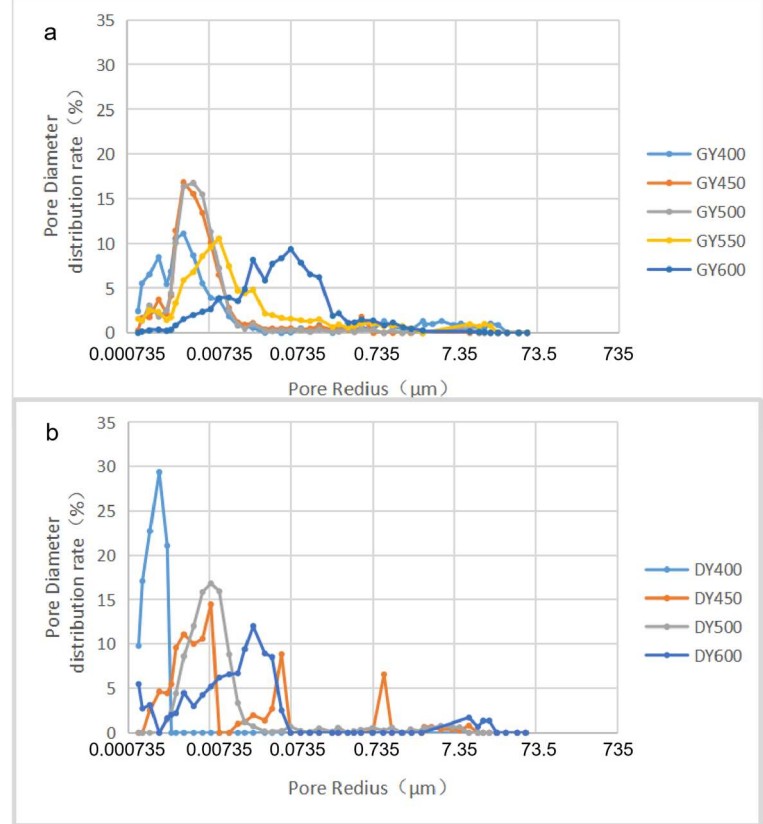

**Figure 9.** Pore size distribution derived from the mercury intrusion experiment. (**a**) Semiclosed and (**b**) open systems.

The nitrogen adsorption and desorption curves (Figure 10) show that the curves of the two systems were in an inverted S shape. The curve shapes of the semiclosed system were basically H3-type loops, while the open-system sample curves were generally H4-type ones. Only the curve of 600 °C depicted characteristics of an H3-type loop. All hysteresis loops were small, indicating complex pore structures and throat sizes. According to the nitrogen adsorption and desorption curves, the effective pores of the samples in the semiclosed system could be mainly wedge-shaped or V-shaped pores with one end or both ends open. In contrast, the open system developed ink bottle-shaped pores with narrow throats. The faster the nitrogen adsorption line rises, the better the pore openness; hence, the pore

structure of the samples in the open system was not as well developed as that in the semiclosed system.

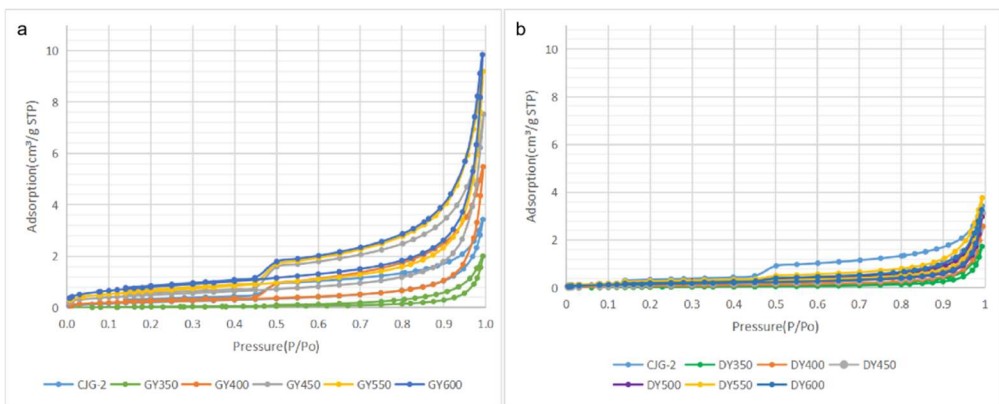

**Figure 10.** Nitrogen adsorption and desorption curves. (**a**) Semiclosed and (**b**) open systems.

The results of the alcohol porosity, nitrogen adsorption specific surface area, and pore volume illustrated that with the maturity increase, the porosity showed an upward trend, and the porosity growth rate of the semiclosed system was much higher than that of the open system (Figure 11a). The differences in the pore volume, pore specific surface area, and pore diameter were quite large. The pore specific surface area and the pore volume of the semiclosed system both depicted the characteristics of "decrease, and then increase" (Figure 11b,c). The pore volume and the specific surface area were the lowest at approximately 1.3% Ro. They then rapidly increased, reaching the maximum at an Ro of approximately 3.5%. Meanwhile, the pore diameter showed a change characteristic of "increase, and then decrease" (Figure 11d) because when the evolution entered the oil generation stage, the liquid hydrocarbons filled the primary pores, and the organic pores were developed during the oil generation, resulting in a decrease in the pore volume and pore specific surface area. The filled pores were reopened due to the secondary cracking of the liquid hydrocarbons in the pores. At the same time, due to the peak period of the hydrocarbon generation, the organic micropores were developed, consequently reducing the average pore size. Therefore, with the increase in maturity, this system was more conducive to the development of the micropores in organic matter. The organic micropores with a better connectivity were more conducive to hydrocarbon storage and migration. For the open system, the pore-specific surface area and the pore volume did not greatly change. They only reached the lowest when Ro was ~1.5%. The pore size showed a trend of "increase, and then decrease," but it was not obvious. The organic matter evolution was restricted by the lack of support from the formation pressure in the open system. Coupled with strong compaction, the pores and the throats were small, and the pore development was generally poor.

### 2.3. Process of Pore Evolution

The thermal evolution experiments under different formation pressures showed that the evolution differences of the pore morphology, distribution, and structure will directly affect the development of the pore system [36–42].

The low-mature stage showed mainly primary inorganic pores, such as brittle mineral intergranular pores, clay mineral interlayer pores, microcracks, and pyrite intercrystalline pores. Only a small amount of organic matter pores was formed due to the biochemistry action and the low-temperature thermal action. The pores depicted a heterogeneous evolution pattern with the increase in simulation temperature.

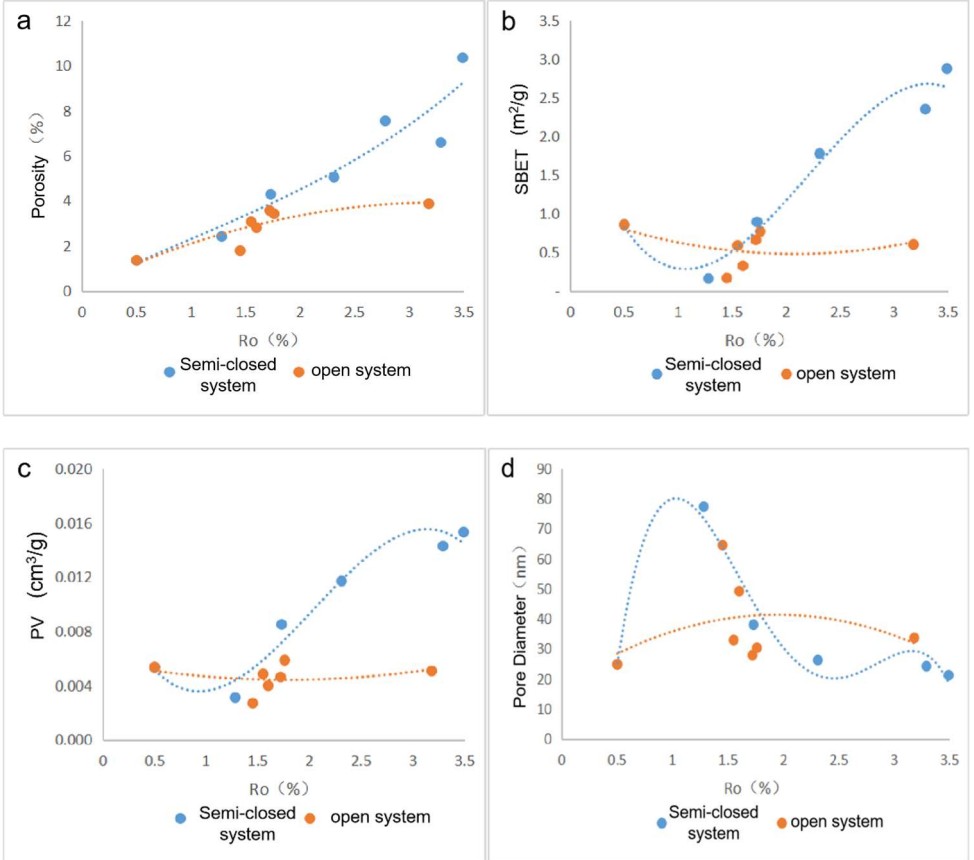

**Figure 11.** Variation trend of the pore structure parameters with maturity. (**a**) Ro-Porosity (**b**) Ro-SBET (**c**) Ro-PV and (**d**) Ro-Pore diameter.

In the semiclosed system, the pore evolution started from low maturity and underwent three stages: mature, high-mature, and overmature:

(1)    Mature stage: At 350–400 °C, the organic matter enters the mature and high-mature stages. The generated hydrocarbon produces a large number of organic pores with a large pore size, and the clay minerals begin to transform into illite. However, in this period, the pores could be filled with liquid hydrocarbons, which will result in pore size reduction and a poor connectivity.

(2)    High-mature stage: At 450–500 °C, the organic matter and the liquid hydrocarbons begin to crack to generate gaseous hydrocarbons. A large number of organic pores are generated inside the organic matter. At this stage, the liquid hydrocarbons filled in the pores are cracked, and the blocked pores are reopened. The mineral dissolution pores in the brittleness around the organic matter begin to develop, and the porosity and the connectivity increase.

(3)    Over mature stage: At 550–600 °C enters gas generation. The macropores are almost invisible. Mesopores and micropores are developed. These pores are important storage space for the hydrocarbon gases generated in the high-mature stage. Influenced by the strong compaction, the pore closure phenomenon will also occur. All these make the pore types of organic matter in the high-mature stage diversified and the pore structure more complex.

In the open system, the organic matter evolution was restricted by the excessive compaction. Therefore, the pore evolution of this system started from low maturity and underwent two stages: high and over maturity.

(1)    High-mature stage: At 350–550 °C, the organic matter is in a high-maturity stage with Ro ≤ 1.8% due to the insufficient formation pressure in the system. Many primary

inorganic pores, such as brittle mineral intergranular pores and clay mineral pores, are observed. With the temperature increase and compaction strengthening, the pore evolution was limited. Only a small number of organic pores and edge cracks was developed.

(2) Overmature stage: At 600 °C, the organic matter maturity enters the overmature stage. The organic matter generates hydrocarbon gas, and a certain formation pressure in the shale promotes the development of the organic matter pores. However, due to excessive compaction, the pore–throat closure is serious, and more ink bottle-shaped pores are formed. The overall pore structure of the organic matter is also poorly developed.

Compared with the open system, the semiclosed system generated more organic pores, cracks, and other well-connected pores. These pores are ideal storage and migration spaces for hydrocarbon gas. On the other side, the semiclosed system enters the overmature stage at 450 °C. This stage with an Ro of 2.0%–3.5% is the best stage for the evolution and development of shale organic pores. This is more conducive to the development of the organic pores and the pore structure of source rocks.

### 3. Conclusions

The following conclusions are obtained from this paper:

(1) The porosity increases nonlinearly with the increase in maturity. The porosity growth rate of the semiclosed system is higher than that of the open system. The semiclosed system mainly develops mesopores, while the open system develops micropores. The macropore development is restricted by the influence of compaction. The key factor influencing the shale pore system evolution is the generation and adjustment of organic pores. The inorganic mineral conversion (kaolinite into illite) contributes less to the pore evolution.

(2) The semiclosed system develops wedge- or V-shaped pores that are open at one or both ends, while the open system develops ink bottle-shaped and closed pores with small throats. The pore openness and the connectivity of the semiclosed system are better than those of the open system.

(3) The semiclosed system is more conducive to the organic pore development than the open system due to the formation pressure. The semiclosed system can reach the overmature stage earlier. The overmature stage is the most important stage for the evolution and development of organic pores.

(4) Using the thermal evolution experiment of the low-maturity shale in Dalong Formation, the evolution process of the high-maturity shale in Longmaxi Formation is forward modeled, guiding the pore evolution of the marine high-maturity shale reservoir in the Sichuan Basin.

**Author Contributions:** Conceptualization, W.Z.; methodology, W.Z.; validation, W.C. and X.Z.; formal analysis, X.Z.; investigation, X.Z. and K.J.; resources, X.Z.; data curation, X.Z.; writing—original draft preparation, X.Z.; writing—review and editing, W.Z., H.X. and W.C.; project administration, W.Z.; funding acquisition, W.Z. All authors have read and agreed to the published version of the manuscript.

**Funding:** The study was supported by National Natural Science Foundation of China (Grant No.: 41972137, 42002157), China Scholarship Council (Grant No.: 202108510120) and National Major Science and Technology Project of China (Grant No.: 2016ZX05034002-006).

**Institutional Review Board Statement:** Not applicable.

**Informed Consent Statement:** Not applicable.

**Data Availability Statement:** Data will be made available upon request.

**Acknowledgments:** Thanks to Wuxi Sinopec Petroleum Exploration and Production Research Institute for their support during the experiment. We appreciate the valuable comments from the editors and anonymous reviewers.

**Conflicts of Interest:** The authors declare no conflict of interest.

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
