# Peer review of "Pore Evolution Characteristics of Marine Organic-Rich Shale Based on a Pyrolysis Simulation Experiment"

_minerals, doi:10.3390/min12091098_

Round 1

Reviewer 1 Report

Although there are many articles about shale,authors thinka semiclosed system with a certain formation pressure is more conducive to the development of organic pores and the pore structure than an open system with a lower formation pressure. The best stage of pore evolution and development is overmature stage.

Paper has innovation,I agree to publish this paper.

Author Response

Thanks to the reviewer for positive affirmation!

Reviewer 2 Report

Generally, the author tried to investigate the pore evolution characteristics of organic-rich shale in semiclosed system and open system, using pyrolysis simulation experiment. Multiple techniques including geochemical and mineralogical measurements, SEM, low temperature N2 adsorption, and mercury intrusion porosimetry (MIP) were used to characterize the change of shale with the temperatures. The pore structure was discussed in detail. The work in this manuscript brings some new and interesting findings. However, there are still some issues that need to be improved. The following queries may be addressed before their final publication.

1. In the abstract section, the author should briefly introduce the problem of your manuscript and its importance in the first sentence. And it could be better if the author improve the description. Such as Line 19 “a semiclosed system with a certain formation pressure…”.  A certain pressure means what? High or low pressure?

2. In the introduction section, there are many new findings in pore structure evolution in recent years. I suggest that the author could cite some new research in recent years and conclude their advantages and disadvantages. Then you can talk what is new in your work.

3. Please check the incorrect use of singular and plural. Such as: Line 38: “ Experiments are an effective approach for...”

4. Some sentences are described in ambiguity or confusion. Such as:

Line 43: “Thermal experiments conducted in laboratories select...”

Lines 54 to 58. “In addition, there are also many imprecisions in the experimental design of thermal evolution. These experiments cannot accurately reflect the pore evolution characteristics and the porosity changes of source rocks in the actual stratigraphic environment.” Please explain more about this argument.

Line 163: “In organic matter with developed pores, the pore development...”

Please check the manuscript words by words.

5. Please check and improve your English grammar. Such as:

Line 287: “Fig. 8 shows that the mercury intrusion curves of the two systems were quite different.”

6. In Fig. 8 Pore size distribution derived from the mercury intrusion experiment, there are some mistakes in the “Pore Radius” axis data. Please check them all.

7. Lines 148 to 149. “The organic matter was distributed between the brittle mineral particles and around the spheroidal pyrite or coexisted with the clay minerals.” Please add picture-related evidence and make notes in the text.

8. Some cited papers in this article are not listed in references, please add them in references. Such as:

Line 283: “The shape of the mercury injection curve can reflect the development of the reservoir pore throats and the pore connectivity (Cao et al., 2016).”

9. In section “3.2 Pore structure”, it might be better if more description of Fig. 7 Mercury intrusion experiment curve of an open system.

10. In the discussion section, the conclusion needs to improve. I suggest that the author could cite others’ results to compare your work and prove your conclusion.

Author Response

Point 1. In the abstract section, the author should briefly introduce the problem of your manuscript and its importance in the first sentence. And it could be better if the author improve the description. Such as Line 19 “a semiclosed system with a certain formation pressure…”.  A certain pressure means what? High or low pressure?

Response 1: A certain pressure means pressure for 15Mpa, higher than that in open system, the pressure is set as 5Mpa.

Point 2. In the introduction section, there are many new findings in pore structure evolution in recent years. I suggest that the author could cite some new research in recent years and conclude their advantages and disadvantages. Then you can talk what is new in your work.

Response 2: I will add new research recent years.

Point 3. Please check the incorrect use of singular and plural. Such as: Line 38: “ Experiments are an effective approach for...”

Response 3: I will check all.

Point 4. Some sentences are described in ambiguity or confusion. Such as:

Line 43: “Thermal experiments conducted in laboratories select...”

Lines 54 to 58. “In addition, there are also many imprecisions in the experimental design of thermal evolution. These experiments cannot accurately reflect the pore evolution characteristics and the porosity changes of source rocks in the actual stratigraphic environment.” Please explain more about this argument.

Line 163: “In organic matter with developed pores, the pore development...”

Please check the manuscript words by words.

Response 4: Some sentences are too long, I will check manuscript soon.

Point 5. Please check and improve your English grammar. Such as:

Line 287: “Fig. 8 shows that the mercury intrusion curves of the two systems were quite different.”

Response 5: I will check and improve English grammer.

Point 6. In Fig. 8 Pore size distribution derived from the mercury intrusion experiment, there are some mistakes in the “Pore Radius” axis data. Please check them all.

Response 6: I will check it soon.

Point 7. Lines 148 to 149. “The organic matter was distributed between the brittle mineral particles and around the spheroidal pyrite or coexisted with the clay minerals.”Please add picture-related evidence and make notes in the text.

Response 7: I will check and note in the text.

Point 8. Some cited papers in this article are not listed in references, please add them in references. Such as:

Line 283: “The shape of the mercury injection curve can reflect the development of the reservoir pore throats and the pore connectivity (Cao et al., 2016).”

Response 8: I have missed some references when I was writing, I will add and list them.

Point 9: In section “3.2 Pore structure”, it might be better if more description of Fig. 7 Mercury intrusion experiment curve of an open system.

Response 9: I will try my best to improve my description.

Point 10: In the discussion section, the conclusion needs to improve. I suggest that the author could cite others’ results to compare your work and prove your conclusion.

Response 10: I will cite more research and improve my conclusion.

Reviewer 3 Report

The authors present an interesting experiment in which they mimic petroleum genesis during (deep) burial of kerogen-bearing shale.  Leaving aside the question of whether pore evolution over the course of two days (lab scale) is representative of pore evolution over two million days (geologic scale), the authors must address two major problems.

1)    Geologic relevance

a.     The description of experimental procedure (ll 112-119 and table 2) is not clear.  There are two possibilities.

                                               i.     Was a sample heated from room temperature to 350 C, kept for 2 days, cooled to room temperature, then heated to 400 C, kept for 2 days, cooled to room temperature, then heated to 450 C, etc?    If so, then the authors must state this clearly, then discuss the effects of cyclic cooling, de-pressuring, then repressuring and reheating on pore evolution.

                                             ii.     Was one sample heated from room temperature to 350 C, kept for 2 days, cooled, and measured, while a different sample was heated from room temperature to 400 C, and a third sample was heated from room temperature to 450 C, etc?  If so then the authors must state this clearly, then explain whether the sample heated rapidly to 400 C from room temperature is representative of the geological situation, in which the sample would be heated slowly from 350 C to 400 C.  The problem gets worse as T increases, because ever more hydrocarbon gets expelled during the two days.  This is likely to exaggerate the pore evolution, compared to letting all the hydrocarbon generated at 350 until formation pressure is balanced, before increasing the temperature to generate additional hydrocarbon.

The key point is that the authors apparently did not carry out the most geologically relevant experiment, which would be to heat a sample to 350, hold for two days, then heat from 350 to 400 and hold for two days, then from 400 to 450 and hold, then 450 to 500 etc.  (The actual protocol would have started with a set of six samples, with one removed for characterization after each increment in temperature).  Without some discussion of how representative the presented experiments are, we are left merely with qualitative evidence of something obvious: more petroleum generation creates more pores. This does not merit publication.

b.     The authors must compare the mercury intrusion and withdrawal curves for at least one of heated samples to curves reported for mature shales in the literature. Numerous examples of the latter have been presented.  This is essential for evaluating the relevance of the experimental maturation process to the natural process.  (Note it does not suffice to compare the inferred pore size distributions, because these leave out essential features of the intrusion/withdrawal curves as discussed below.)

c.     The authors must include the mercury curves for the unheated sample. Knowing the initial state (prior to heating) is essential for understanding the heated samples in Figure 7. 

2)    Pore structure and connectivity interpretation.

a.     The analysis of the mercury intrusion and withdrawal curves in Figure 7 is inadequate.  Numerous key features have not been recognized or discussed. There is no systematic trend as temperature increases, so the statement “overall upward trend” (l 306) makes no sense.

                                               i.     It is clear that sample GY400 has significant fraction of pore space accessible between 0.05 and 5 MPa.  But none of this pore space is accessible for the higher temperature samples GY450 and GY500.  Why not?

                                             ii.     On the other hand samples GY-400, -450 and -500 all have pore space accessed between 50 and 500 MPa. But that pore space constitutes widely different fractions of the total pore space in each sample.  Why?

                                            iii.     The fraction of gas-accessible pore space that is accessible to mercury at 500 MPa varies widely (38% for GY450; >95% for GY500 and 600) and nonmonotonically with temperature.  Why?

                                            iv.     The withdrawal curves show very different amounts of retained mercury: almost all the intruded mercury was retained in GY400 and 450, but less than 40% of it in GY600.  Why?

b.     The underlying but not discussed phenomenon is pore space connectivity. The authors’ experiments nicely show that petroleum generation creates its own pore space and hydrocarbon expulsion creates its own connected paths.  But the mercury curves show remarkably unsystematic variation in connectivity as thermal maturity increases.   Surely continued petroleum generation as maturation continues would monotonically increase pore space connectivity?  The authors must address why they think their observations are representative of shales that undergo natural thermal maturation.

There are other points to be addressed that would improve the paper:

1)    How much brine was expelled at temperature? Thermal expansion of the brine phase could have been significant influence on pore evolution. This would have started well before petroleum generation. 

2)    How much water vapor was produced? Was there sufficient evaporation to precipitate salts?

3)    Pore evolution in the open conditions was much less than semi-closed conditions.  Does this mean that the compression of OM is a significant contribution to pore evolution during burial?  This was not adequately discussed.

4)    How much hydrocarbon was generated? How much of the created porosity (Fig 10) corresponds to that volume? The TOC in Table 2 have similar values that are much less than the initial value of TOC in Table 1, suggesting that all the heated samples should have similar porosities much larger than the initial value.  Fig 10 contradicts this expectation. The authors should account for this.  

5)    Figure 8 is poor quality.  The important features are hidden by the large filled symbols.

Author Response

Dear Editor,

Thank you very much for your letter, as well as constructive comments from the referees, about our manuscript “Pore evolution characteristics of marine organic-rich shale based on a pyrolysis simulation experiment”. We have carefully revised it according to the comments. Here, we attach a revised manuscript in the format of MS Word for your approval; changes in the revised manuscript are marked in red for easy tracking.

Response to Reviewer #3:

Comment 1. The description of experimental procedure (ll 112-119 and table 2) is not clear.There are two possibilities.

i.Was a sample heated from room temperature to 350 C, kept for 2 days, cooled to room temperature, then heated to 400 C, kept for 2 days, cooled to room temperature, then heated to 450 C, etc? If so, then the authors must state this clearly, then discuss the effects of cyclic cooling, de-pressuring, then repressuring and reheating on pore evolution.

  1. Was one sample heated from room temperature to 350 C, kept for 2 days, cooled, and measured, while a different sample was heated from room temperature to 400 C, and a third sample was heated from room temperature to 450 C, etc?  If so then the authors must state this clearly, then explain whether the sample heated rapidly to 400 C from room temperature is representative of the geological situation, in which the sample would be heated slowly from 350 C to 400 C. The problem gets worse as T increases, because ever more hydrocarbon gets expelled during the two days. This is likely to exaggerate the pore evolution, compared to letting all the hydrocarbon generated at 350 until formation pressure is balanced, before increasing the temperature to generate additional hydrocarbon.

The key point is that the authors apparently did not carry out the most geologically relevant experiment, which would be to heat a sample to 350, hold for two days, then heat from 350 to 400 and hold for two days, then from 400 to 450 and hold, then 450 to 500 etc.  (The actual protocol would have started with a set of six samples, with one removed for characterization after each increment in temperature). Without some discussion of how representative the presented experiments are, we are left merely with qualitative evidence of something obvious: more petroleum generation creates more pores. This does not merit publication.

Response 1 : Thanks for your critical comments. As mentioned, we have re-phrased the relevant section to make it clear. In the experiment, each time heat only one sample, and more details about the experimental procedure has been added and present in the Section 1.1: The first sample heated from room temperature to 350 C, and kept for 2 days; cooled, and take it out from sample chamber and measured; then heated the second sample to 400 C, kept for 2 days; cooled and take it out and measured; then heat the third sample to 450 C, etc.

The heat rate was set to 20 C/h and the hydrocarbons are expelled through a pipeline connected to the sample chamber, in order to keep system pressure in 15Mpa for semiclosed system and 5Mpa for open system, the additional hydrocarbon have not influenced the pore evolution.

Comment 2. The authors must compare the mercury intrusion and withdrawal curves for at least one of heated samples to curves reported for mature shales in the literature. Numerous examples of the latter have been presented. This is essential for evaluating the relevance of the experimental maturation process to the natural process. (Note it does not suffice to compare the inferred pore size distributions, because these leave out essential features of the intrusion/withdrawal curves as discussed below.)

Response 2: This excellent comment has been addressed in our revised manuscript. We compare the mercury intrusion curves of the heated samples and the Longmaxi Formation shale samples, the results show that the curves have some similarities, the heated samples and natural thermal mature samples are similar in pore development, pore shape and pore connectivity.

Comment 3. The authors must include the mercury curves for the unheated sample. Knowing the initial state (prior to heating) is essential for understanding the heated samples in Figure 7.

Response 3: I have do the mercury intrusion experiment for unheated sample, but it was failed to form a curve because of the poor porosity. So there are only heated samples’ curves in Fighre 7.

Comment 4: The analysis of the mercury intrusion and withdrawal curves in Figure 7 is inadequate.  Numerous key features have not been recognized or discussed. There is no systematic trend as temperature increases, so the statement“overall upward trend”(l 306) makes no sense.

  1. It is clear that sample GY400 has significant fraction of pore space accessible between 0.05 and 5 MPa. But none of this pore space is accessible for the higher temperature samples GY450 and GY500. Why not?

Response 4i: It is speculated that in GY400, the temperature reaches 400 C, organic matter cracking to produce oil, and produce organic pores and inorganic pores with larger pore diameter, then in GY450, the temperature reaches 450 C, the oil block the pore space and cause the pore diameter smaller; in GY500, the temperature reaches 500 C, the organic matter and the oil in the pores begin to crack into hydrocarbon gases, produce a great amount of micropores, and with the increased pressure, lead to smaller pore diameter.

  1. On the other hand samples GY-400, -450 and -500 all have pore space accessed between 50 and 500 MPa. But that pore space constitutes widely different fractions of the total pore space in each sample. Why?

Response 4ii: Actually, the samples in the experiment were not the same one. Because of the heterogeneity of the shale sample, the distribution of mineral components and organic matters are different, lead to the difference distribution of pores and different fractions of the pore space.

  • The fraction of gas-accessible pore space that is accessible to mercury at 500 MPa varies widely (38% for GY450; >95% for GY500 and 600) and nonmonotonically with temperature.  Why?

Response 4iii: Mercury curve data cannot reflect the pore connectivity. We use mercury curve data to analyze the micropore development of samples at different thermal evolution stages. Because even if the organic matter and mineral components are equivalent, due to their different distribution, the micropore distributions are different, the curve characteristics are also different.

  1. The withdrawal curves show very different amounts of retained mercury: almost all the intruded mercury was retained in GY400 and 450, but less than 40% of it in GY600. Why?

Response 4iv: It is speculated that in temperature 400 C, the organic matter start cracking to produce oil, and oil black the pore space in 450 C, when temperature reaches 600 C, the oil and organic matter crack to hydrocarbon gases. These factors indicate GY600 has well developed pores and throats, and pore connectivity is better than GY400 and 450, lead to a lower retained mercury saturation.

Comment 5: The underlying but not discussed phenomenon is pore space connectivity. The authors’ experiments nicely show that petroleum generation creates its own pore space and hydrocarbon expulsion creates its own connected paths. But the mercury curves show remarkably unsystematic variation in connectivity as thermal maturity increases. Surely continued petroleum generation as maturation continues would monotonically increase pore space connectivity? The authors must address why they think their observations are representative of shales that undergo natural thermal maturation.

Response 5: Thanks for your critical comments which lead us to a lot of thinking. Pore space connectivity are important, but the main propose of this manuscript is to reveal the pore evolution of organic pores, inorganic pores in different evolution stage and their developmental patterns. It is also noted that the 3D SEM images, instead of 2D SEM images, are usually applied to characterize the complex pore connectivity. In addition, we have compared the micropore development through the mercury injection curve of the heated samples and the natural thermal maturation samples, to simulate the geological shales that undergo natural thermal maturation.

Round 2

Reviewer 3 Report

the responses to 1 and 2 are satisfactory.

the responses to 3, 4 and 5 introduce important additional information that readers must have in order to properly understand the basis for the authors' assertions and conclusions.  The information in these responses must be included in the manuscript.  

Author Response

Thanks to the reviewer for comments and positive affirmation!These suggestions will be revised and supplemented in the manuscript.